

# CRISPR/Cas9-mediated deletion of the Wiskott-Aldrich syndrome locus causes actin cytoskeleton disorganization in murine erythroleukemia cells

Vanessa Fernández-Calleja[1], María-José Fernández-Nestosa[2], Pablo Hernández[1], Jorge B. Schvartzman[1] and Dora B. Krimer[1]

[1] Department of Cellular and Molecular Biology, Centro de Investigaciones Biológicas, Spanish National Research Council (CSIC), Madrid, Spain
[2] Bioinformatic Laboratory, Polytechnic School, National University of Asuncion, San Lorenzo, Paraguay

## ABSTRACT

Wiskott-Aldrich syndrome (WAS) is a recessive X-linked inmmunodeficiency caused by loss-of-function mutations in the gene encoding the WAS protein (WASp). WASp plays an important role in the polymerization of the actin cytoskeleton in hematopoietic cells through activation of the Arp2/3 complex. In a previous study, we found that actin cytoskeleton proteins, including WASp, were silenced in murine erythroleukemia cells defective in differentiation. Here, we designed a CRISPR/Cas9 strategy to delete a 9.5-kb genomic region encompassing the *Was* gene in the X chromosome of murine erythroleukemia (MEL) cells. We show that *Was*-deficient MEL cells have a poor organization of the actin cytoskeleton that can be recovered by restoring *Was* expression. We found that whereas the total amount of actin protein was similar between wild-type and *Was* knockout MEL cells, the latter exhibited an altered ratio of monomeric G-actin to polymeric F-actin. We also demonstrate that *Was* overexpression can mediate the activation of Bruton's tyrosine kinase. Overall, these findings support the role of WASp as a key regulator of F-actin in erythroid cells.

Corresponding author
Dora B. Krimer, dbkrimer@cib.csic.es

## INTRODUCTION

Wiskott-Aldrich syndrome (WAS) is an X-linked hematological disorder clinically characterized by microthrombocytopenia, eczema, recurrent infections and predisposition to develop lymphomas and autoimmunity diseases (*Massaad, Ramesh & Geha, 2013*; *Matalon, Reicher & Barda-Saad, 2013*; *Thrasher & Burns, 2010*). The disease arises from mutations in the gene encoding the WAS protein (WASp), resulting in cytoskeletal abnormalities that are responsible for the wide spectrum of clinical phenotypes. The levels of WASp expression correlate negatively with the severity of the disease; accordingly, low levels produce milder forms such as X-linked thrombocytopenia, whereas the absence of the protein results in the most serious manifestations of Wiskott-Aldrich syndrome (*Albert et al., 2010*; *Jin et al., 2004*; *Zhu et al., 1997*).

WASp is the founder member of a family of actin nucleation-promoting factors that includes at least five subfamilies, WASP (WASP and neuronal N-WASP), SCAR/WAVE (verprolin homolog isoforms), WASH (WASP and SCAR homolog isoforms), WHAMM (WASP homolog associated with actin, membranes and microtubules), and JMY (junction-mediating regulatory protein) (*Alekhina, Burstein & Billadeau, 2017*), and all are important regulators of actin cytoskeletal dynamics. WASp family proteins function as multidomain proteins that adopt a closed autoinhibited conformation, where the carboxy-terminal verprolin-cofilin-acidic domain interacts with the GTPase-binding domain. Binding of the Rho GTPase Cdc42 protein unlocks the verprolin-cofilin-acidic region, allowing binding and activation of the Arp2/3 (actin-related protein) complex, which subsequently stimulates actin polymerization (*Kim et al., 2000*). As opposed to the ubiquitous expression of other WASp family members, WASp is exclusively expressed in the hematopoietic lineage (*Derry, Ochs & Francke, 1994*; *Parolini et al., 1997*), including erythroid cells (*Fernandez-Calleja et al., 2017*; *Parolini et al., 1997*),

The eukaryotic actin cytoskeleton is a key element in various cellular processes and is classically associated with cell migration, adhesion, endo/exocytosis, and cytokinesis. Actin microfilaments are composed of monomeric globular (G-actin) and polymeric, filamentous actin (F-actin), and the dynamic transition between the two forms are dependent on the cellular requirements (*Rotty & Bear, 2014*). Increasing data show that actin dynamics is also involved in signaling regulation and plays important roles in cell differentiation (*Misu, Takebayashi & Miyamoto, 2017*) and tumor progression (*Ebata, Hirata & Kawauchi, 2016*; *Nurnberg, Kitzing & Grosse, 2011*). We recently demonstrated that the actin cytoskeleton is poorly organized in a murine erythroleukemia (MEL) cell line resistant to cell differentiation (*Fernandez-Calleja et al., 2017*). Unlike the progenitor MEL-DS19 cell line, resistant cells (MEL-R) are refractive to most inducers capable of triggering cell differentiation (*Fernandez-Nestosa et al., 2008*; *Fernandez-Nestosa et al., 2013*). Transcriptome profiling by next-generation sequencing of MEL-DS19 and MEL-R cell lines revealed that several genes involved in actin polymerization were poorly expressed in MEL-R cells. WASp emerged as one of the actin-related proteins whose expression was blunted in resistant cells (*Fernandez-Calleja et al., 2017*). Accordingly, MEL-R cells showed a marked decrease in actin content as measured by immunocytochemistry and confocal microscopy, even though the total amount of actin protein remained unchanged. Based on these observations, we suggested that actin-related proteins might shape the cytoskeleton organization. Furthermore, we hypothesized that the loss of any of the actin-network components may interfere with the dynamic assembly that takes place during actin polymerization. In the present study, we used CRISPR/Cas9 to delete the *Was* locus in MEL DS19 cells. We found that loss of WASp altered the dynamics of filamentous actin (F-actin) and free globular actin (G-actin) turnover, which led to an aberrant actin cytoskeleton organization. The phenotype displayed by the CRISPR/Cas9-edited *Was* transfectants resembled that of MEL-R cells, and could be recovered by WASp overexpression. We also show that ectopic expression of WASp enhances the expression of Bruton's tyrosine kinase, an important component of the actin cytoskeleton network.

## MATERIALS AND METHODS

### Cell cultures

The murine erythroleukemia cell line MEL-DS19 (hereafter called MEL) was obtained from Dr Arthur Skoultchi (Albert Einstein College of Medicine, Bronx, New York, USA). MEL-R cells, derived from MEL cells, were previously established in our laboratory by growing MEL cells continuously in the presence of 5 mM hexamethylene bisacetamide (HMBA) (*Fernandez-Nestosa et al., 2008*; *Fernandez-Nestosa et al., 2013*). Murine 3T3-Swiss albino fibroblasts (CCL-92) were obtained from the ATTC. Cell lines were propagated in Dulbecco's Modified Eagle's Medium containing 10% fetal bovine serum (BSA), 100 units/ml penicillin and 100 µg/ml streptomycin (all from Gibco). MEL-R cells were routinely cultured in the presence of 5 mM HMBA (Sigma). MEL DS19 cell differentiation was induced by exposing exponentially growing cultures to 5 mM HMBA. Hemoglobinized cells were quantified by determining the proportion of benzidine-staining positive cells (B+) in cell cultures. Cell growth was assessed daily by counting samples of the cultures with a Neubauer hemocytometer chamber.

### Generation of MEL/*Was*$^{-/-}$ cells by CRISPR/Cas9 technology

Genomic deletion of *Was* in MEL cells was performed by CRISPR/Cas9 technology as described (*Bauer, Canver & Orkin, 2015*). Two single guide RNAs (sgRNA1 and sgRNA2) were designed to separately target the entire gene (mouse X:7658591–7667617) using the online tool CRISPR (http://crispr.mit.edu/). Coupled complementary oligonucleotides (CACC was added to the 5′ end of the sense strand and AAAC was added to the 5′ end of the antisense strand) were annealed and inserted into the BbsI sites of linearized pX330 vector (Addgene plasmid ID 42230). The sequences of the sgRNA oligonucleotides are listed in Table S1. MEL cells were co-transfected with the two sgRNAs vectors and a third vector, pEFBOS-GFP, encoding green fluorescent protein (GFP) (Fig. S1), by cationic liposome-based transfection with Lipofectamine 2000 (Life Technologies). After 72 h, the top ∼3% of GFP-positive cells were individually sorted into 96-well plates. Genomic DNA was isolated from all clones and screened for biallelic deletion *via* PCR using non-deletion (ND) primers, whose assembly takes place internal to the sequence to be deleted, and deletion (D) primers, whose assembly is upstream and downstream of the sgRNA cleavage sites. The primers used for identifying biallelic deletion clones are listed in Table S2 and were designed with Primer3 software (http://bioinfo.ut.ee/primer3-0.4.0/) (*Untergasser et al., 2012*). The conditions for the PCR were as follows: pre-denaturing step of 94 °C for 7 min, followed by 35 cycles of 94 °C for 40 s, 60 °C for 1 min and 72 °C for 1 min, with a final extension at 72 °C for 7 min. PCR products were resolved on 1% agarose gels and visualized by ethidium bromide staining.

### MEL-R DNA transfection

Exponentially growing MEL-R cells were transfected as described above with the pcDNA3.1 ± DYK *Was* expression vector (GeneScript) containing the coding region of *Was* (hereafter called pcDNA3.1-*Was*) (Fig. S1). After 6 h, cells were distributed into 96-well plates.

The transfectants were selected by limited dilution and maintained in growth medium containing 1 µg/ml G418 (Sigma).

## Antibodies and immunoblotting

Control 3T3 fibroblast cells, MEL, MEL-R and transfected cells ($2.5 \times 10^6$) were harvested, washed with phosphate buffered saline (PBS) and lysed with NP-40 buffer (20 mM Tris-HCl pH 7.5, 10% glycerol, 137 mM NaCl, 1% NP-40, 1 mM sodium orthovanadate, 10 mM sodium fluoride, and 2 mM EDTA) containing protease inhibitors (all from Sigma). Protein lysates (10–30 µg) were separated by 12% SDS-polyacrylamide gel electrophoresis and transferred to PVDF membranes (Bio-Rad). The membranes were incubated with a mouse monoclonal anti-β-actin (1:10,000; Sigma), mouse monoclonal anti-WASp (1:500, Santa Cruz), mouse monoclonal anti-Btk (1:500; Santa Cruz), and rabbit polyclonal anti-α-tubulin (1:1000; ABclonal) antibodies, then washed five times with T-TBS (20 mM Tris–HCl, 150 mM NaCl, 0.1% Tween 20). Primary antibodies were detected with horseradish peroxidase-conjugated anti-mouse (1:3,000; Santa Cruz) or anti-rabbit IgG (1:1,000, DAKO) antibodies, followed by five cycles of T-TBS washes. The analysis of filamentous (F-actin) and globular (G-actin) actin content was performed from $10^7$ cells. Samples were harvested, washed in PBS and lysed in a lysis buffer (50 mM PIPES pH 6.9 (Sigma), 5 mM MgCl$_2$, 5 mM EGTA (Sigma), 5% glycerol (Roche), 0.1% β-mercaptoethanol (Merck), 1 mM PMSF (Roche), 10 mM benzamidine (Sigma) and 1 mM ATP (Roche)). Protein supernatants and pellets were collected after ultracentrifugation (100,000 g, 1 h at 37 °C) and analyzed by immunoblotting as described above.

## Immunofluorescence staining and confocal microscopy

MEL, MEL-R and transfected cells were plated on poly-L-lysine-coated slides and incubated at 37 °C for 30 min. Cells were fixed with 4% paraformaldehyde for 30 min, permeabilized with 0.1% Triton-X 100 in PBS for 30 min and blocked with 1% BSA in PBS/0.1% Triton-X 100 for 1 h, all at room temperature (RT, ~22 °C). Cells were stained with anti-β-actin (1:3,000; Sigma) or anti-Btk (1:200; Santa Cruz) antibodies for 1 h at RT followed by washing twice with PBS. Primary antibodies were detected with an Alexa Fluor 568 secondary antibody (Molecular Probes) and 1 µg/ml DAPI (4,6-diamidino-2-phenylindole; Sigma) was added to stain nuclei, for 1 h at RT, followed by two washes with PBS. Finally, cells were mounted on cover slips with Prolong Diamond Antifade Mountant reagent (Invitrogen). Fluorescence images were acquired on a Leica TCS SP2 confocal microscope using a 100× objective with zoom.

## Statistical analysis

Data are presented as means ± standard deviation of the densitometric analysis. Differences were tested by the Student $t$-test. The values $P < 0.05$ were considered statistically significant. Statistical analysis of western blot data and immunofluorescence images are presented in Figs. S2 and S3, respectively.
## RESULTS

### Effects of ectopic WASp expression on the actin cytoskeleton in MEL-R cells

It is generally recognized that WASp plays an important role in the maintenance of cytoskeletal organization in hematopoietic cells (*Alekhina, Burstein & Billadeau, 2017*). In our previous study, we found that *Was* is expressed in erythroleukemia MEL cells, whereas it was barely detectable in MEL-R cell lines (*Fernandez-Calleja et al., 2017*). To further evaluate the status of *Was* at the protein level, we compared proteins extracts from MEL cells, both undifferentiated and differentiated in the presence of the inducer-mediated differentiation HMBA, with those from a representative MEL-R resistant line. We also included the 3T3 fibroblast cell line as a negative control as WASp is not expressed in non-hematopoietic lineages. Immunoblotting showed robust WASp expression in MEL cells, which decreased during cell differentiation (Fig. 1A). By contrast, the levels of WASp in MEL-R cells were much lower than in undifferentiated MEL cells and were comparable with that observed in 3T3 fibroblasts. Because our recent study (*Fernandez-Calleja et al., 2017*) suggested that low levels of several actin-cytoskeletal proteins, including WASp, result in anomalous cytoskeleton organization, we sought to determine whether forced expression of WASp might reverse this effect. To do this, we established MEL-R cell lines constitutively expressing WASp and screened for clones with robust WASp expression, which identified clones 9, 10 and 11 for further analysis (Fig. 1B).

Confocal immunofluorescence analysis of MEL cells stained with an anti-actin antibody revealed a clear rim of actin fluorescence surrounding the nuclear periphery (Fig. 2, column 1). A similar pattern was observed in the WASp-overexpressing MEL-R clones 9, 10 and 11 (Fig. 2, columns 2–4), whereas no detectable signal was evident in non-Wasp-expressing MEL-R cells (Fig. 2, column 5). These results confirm that WASp expression has a positive impact on the organization of the actin cytoskeleton in erythroleukemia cells.

We next used immunoblotting to evaluate total actin protein levels in MEL-R and WASp-overexpressing MEL-R transfectants, finding no differences between the two (Fig. 3A). Given this result, we next asked whether the changes observed by confocal microscopy might be due to an altered ratio of monomeric G-actin and polymeric F-actin. To address this, we used high-speed centrifugation to separate G- and F-actin pools from cell lysates, followed by immunoblotting with an anti-β actin antibody. Results showed that the F-actin pool in WASp-overexpressing MEL-R clones 9, 10 and 11 was considerably greater than in MEL-R cells, whereas the free G-actin pool remained low (Fig. 3B). By contrast, the G- and F-actin pools in control MEL-R cells were similar. These results demonstrate that the proportion of F-actin increases after WASp overexpression and suggest that it might contribute to nuclear actin polymerization.

### CRISPR/Cas9-mediated *Was* deletion in MEL cells

Proteins associated with the actin polymerization machinery, such as WASp, are silenced or minimally expressed in MEL-R cells (*Fernandez-Calleja et al., 2017*). We have shown that ectopic expression of WASp in MEL-R cells can restore the wild-type phenotype observed for the actin cytoskeleton (Fig. 2). Assuming that WASp is essential for actin

**A**

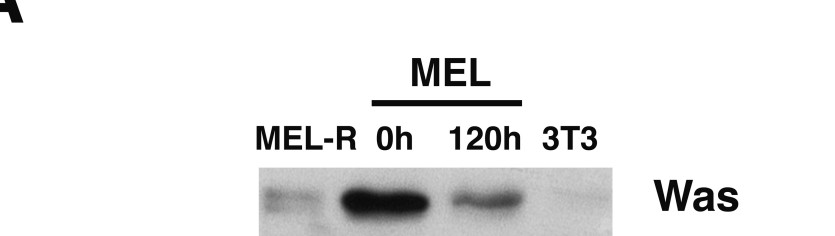

**B**

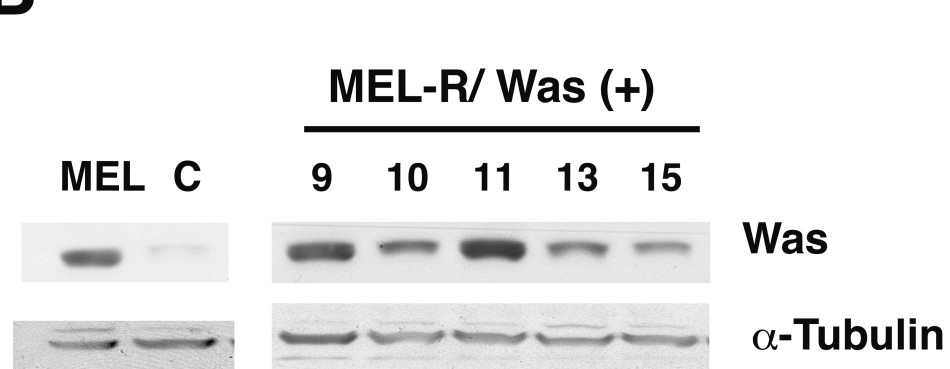

**Figure 1** **WASp is poorly expressed in MEL-R cells.** (A) Immunoblot analysis of whole-cell extracts from erythroleukemia-resistant cells (MEL-R), MEL cells undifferentiated (0 h) and differentiated with hexamethylene bisacetamide (HMBA) (120 h), and 3T3 fibroblasts. Equal amounts of protein (30 μg) were fractionated by SDS-polyacrylamide gel electrophoresis and analyzed by immunoblotting with an anti-Was antibody. $\alpha$-tubulin was used as a loading control. (B) Immunoblot analysis of whole-cell lysates from stable transfectants overexpressing *Was* (MEL-R/*Was* (+) processed as in A). Numbers above the panel correspond to clones 8, 9, 10, 11, 13 and 15. MEL cells and MEL-R cells transfected with an empty vector (C) were treated and analyzed under similar conditions. $\alpha$-tubulin was used as a loading control.

polymerization in erythroleukemia cells, we hypothesized that the knockout of the gene would alter actin cytoskeleton organization in MEL cells. Thus, we used the CRISPR/Cas9 gene-editing platform to delete the genomic region encompassing the *Was* gene in the X chromosome of MEL cells. A schematic representation of the genomic target sites is shown in Fig. 4A. Cells were transfected with three plasmids: pX330*Was* 1 and pX330*Was* 2, expressing the sgRNAs, and pEFBOS-GFP (Fig. S1). Cells were then double selected for GFP-positivity and G418 resistance. PCR analysis of bi-allelic GFP-positive cells (Fig. 4B) and immunoblotting (Fig. 4C) confirmed three knockout clones: 1, 4 and 73. Cell clone-1 was used in most of the following experiments.

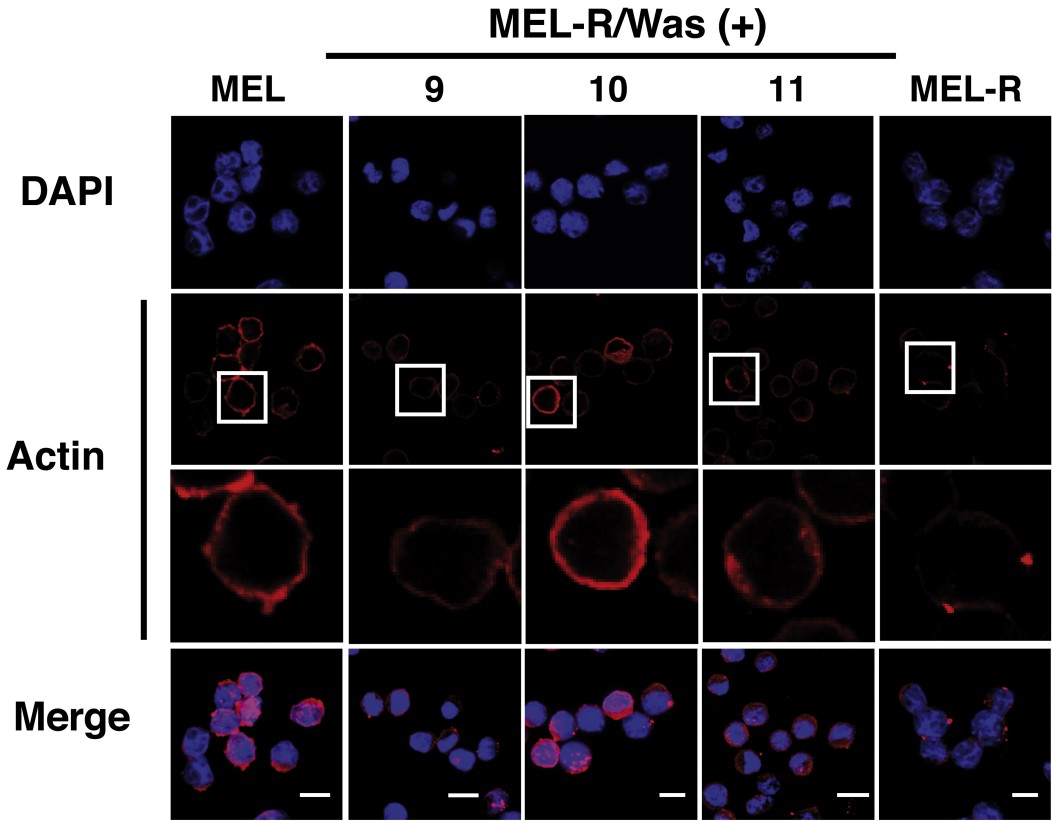

**MEL-R/Was (+)**

MEL | 9 | 10 | 11 | MEL-R

DAPI

Actin

Merge

**Figure 2** **Overexpression of *Was.* induces organization and polymerization of actin cytoskeleton in MEL-R cells.** Immunofluorescence staining of MEL cells, MEL-R/*Was* (+) transfectants 9, 10 and 11, and MEL-R cells with a mouse monoclonal anti-b-actin antibody (red). Nuclei were visualized with DAPI (blue). Magnified views indicated by white boxed areas are shown below second-row panels. The scale bar represents 10 mm.

We first assessed whether *Was* knockout (MEL/*Was*$^{-/-}$) cells had any noticeable phenotypic differences compared with wild-type MEL cells. Cell proliferation analysis showed no significant alterations in growth rate between MEL and MEL/*Was*$^{-/-}$ cells (Fig. S4A ). Cell differentiation was also evaluated after treatment with 5 mM HMBA, a potent inducer of cell differentiation in erythroleukemia cells (Fig. S4B ). Results showed no changes in the percentage of cell differentiation between the two cultures, indicating that CRISPR/Cas9-mediated deletion had no deleterious effects on cell transfectants.

We then questioned whether MEL/*Was*$^{-/-}$ cells presented a defect in the actin cytoskeleton organization. By confocal analysis, loss of *Was* function in MEL/*Was*$^{-/-}$ cells led to a considerably reduced actin fluorescence signal, which was similar in intensity to that observed in MEL-R cells (Fig. 5 compare columns 2 and 4). Since the ectopic overexpression of *Was* in MEL-R cells could rescue the wild-type phenotype (Fig. 2), we performed a similar analysis in MEL/*Was*$^{-/-}$ cells. We transiently transfected the pcDNA3.1-*Was* vector into MEL/*Was*$^{-/-}$ cells and collected cells 48 h later for immunofluorescence analysis. As expected, a rim of actin staining was clearly visible around the nuclei of MEL/*Was*$^{-/-}$ cells

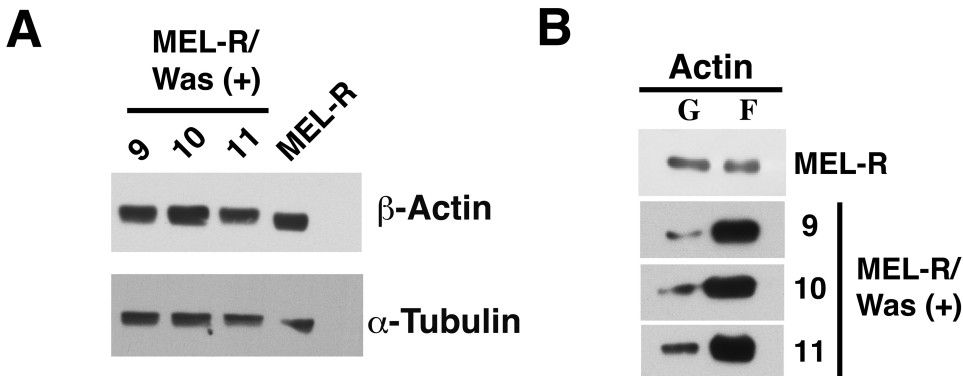

**Figure 3** **Overexpression of *Was* enhances the formation of F-actin in MEL-R transfectants.** (A) Total actin expression was evaluated in MEL-R/*Was* (+) clones 9, 10 and 11, and MEL-R cells by immunoblotting with an antibody against b-actin. $\alpha$-tubulin was used as a loading control. (B) G-actin and F-actin from MEL-R and MEL-R/*Was* (+) transfectants 9, 10 and 11, separated after ultracentrifugation (G-actin remains in the supernatant, F-actin found in the pellet) were immunoblotted and probed as in (A).

overexpressing *Was*, and was comparable with that observed in MEL cells (Fig. 5, compare columns 3 and 1). Overall, the knockout and rescue experiments confirm the important effect of WASp for the actin cytoskeleton organization in erythroleukemia cells.

To examine whether changes in actin expression or an altered ratio of monomeric G-actin and polymeric F-actin was the origin of the actin phenotype in MEL-$Was^{-/-}$ cells, we first measured the expression of actin in MEL-$Was^{-/-}$ cells by immunoblotting We found that the total amount of actin was similar to that in MEL and MEL-R cell lines (Fig. 6A). We then used ultracentrifugation to fractionate monomeric G-actin to the supernatant and polymeric F-actin to the pellet from MEL-$Was^{-/-}$ whole extracts and immunoblotted these fractions against actin. Results showed that the balance between G- and F-actin shifted towards G-actin, with markedly less F-actin detected in the pellet of the *Was* mutant (Fig. 6B).

## Ectopic expression of WASp promotes Btk activation in MEL-R and MEL/$Was^{-/-}$ cells

There is increasing evidence that actin is also present in the cell nucleus, which is referred to as "nuclear actin", where it participates in transcriptional activation and chromatin remodeling (for a recent review see *Misu, Takebayashi & Miyamoto, 2017*). Sadhukan and co-workers previously demonstrated a nuclear role for WASp in transcriptional activation of several master genes during Th1 cell differentiation, a role that is independent of actin polymerization (*Sadhukhan et al., 2014*). In a different study, WASp was reported to be present both in the cytoplasm and the nucleus, and regulated gene transcription in K562 myeloid cells (*Looi et al., 2014*). Because WASp can physically associate with Bruton's tyrosine kinase (Btk) (*Baba et al., 1999*; *Sakuma et al., 2012*), and since Btk was included among the silenced actin-associated proteins in MEL-R cells (*Fernandez-Calleja et al., 2017*), we finally asked whether the ectopic expression of *Was* affects Btk expression.

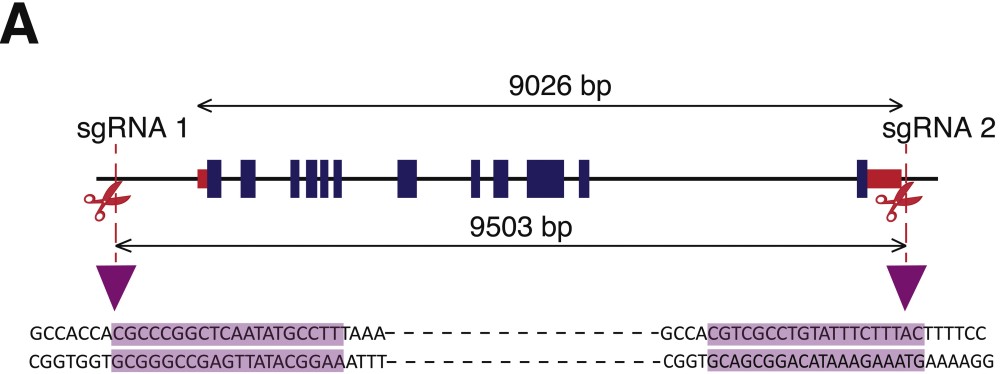

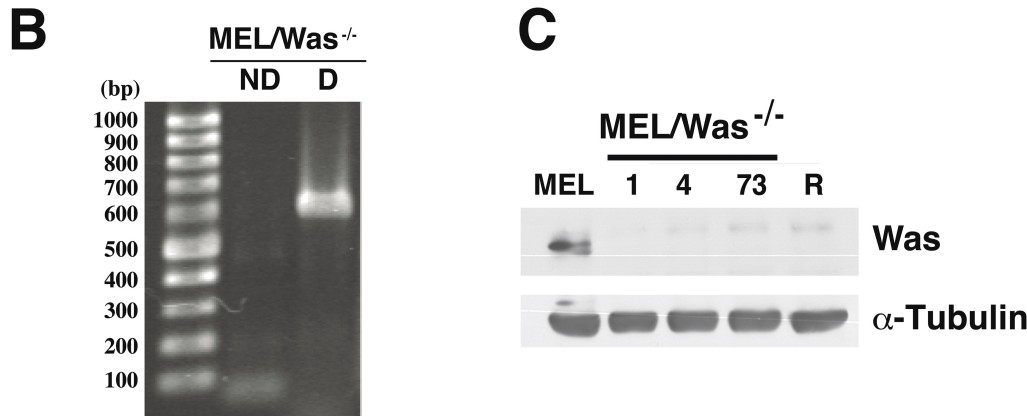

**Figure 4 Deletion of *Was* in MEL cells using CRISPR/Cas9.** (A) Genomic map of *Was* in mouse chromosome X:7658591–7667617, including exons (blue rectangles) and 5′ and 3′ untranslated regions (red rectangles). sgRNA positions in the genome are shown as vertical discontinued red lines. The sgRNA sequences are highlighted in purple and illustrate the Cas9 cleavage region. (B) PCR analysis for screening biallelic deletion clones using primers listed in Fig. S3. PCR products (for clone 1) of the non-deletion amplicon (ND) and the deletion amplicon (D) were electrophoresed on a 1% agarose gel and stained with ethidium bromide. (C) Immunoblotting of total lysates from MEL, MEL/Was$^{-/-}$ clones 1, 4 and 73, and MEL-R cells. $\alpha$-tubulin was used as a protein loading control.

Figure 7 shows the results of immunoblotting of whole-cell lysates from the stable WASp-overexpressing MEL-R transfectants 9, 10 and 11 against an anti-Btk antibody. We observed that Btk was expressed at higher levels in all three transfectants than in MEL-R cells, although it was more evident in clones 10 and 11. We also analyzed MEL-*Was*$^{-/-}$ cells (clones 1 and 73), before and after enforced expression of WASp. In all cases, Btk was expressed at a level similar to that of the MEL progenitor line irrespective of whether Wasp was overexpressed or not (Fig. 7). These results reveal that *Was* overexpression can mediate Btk activation in MEL-R cells, suggesting that Btk is downstream of *Was* in the signaling cascade. By contrast, an alternative signaling pathway might be operative in MEL cells with targeted deletion of *Was*. Indeed, Btk expression can be modulated by different upstream

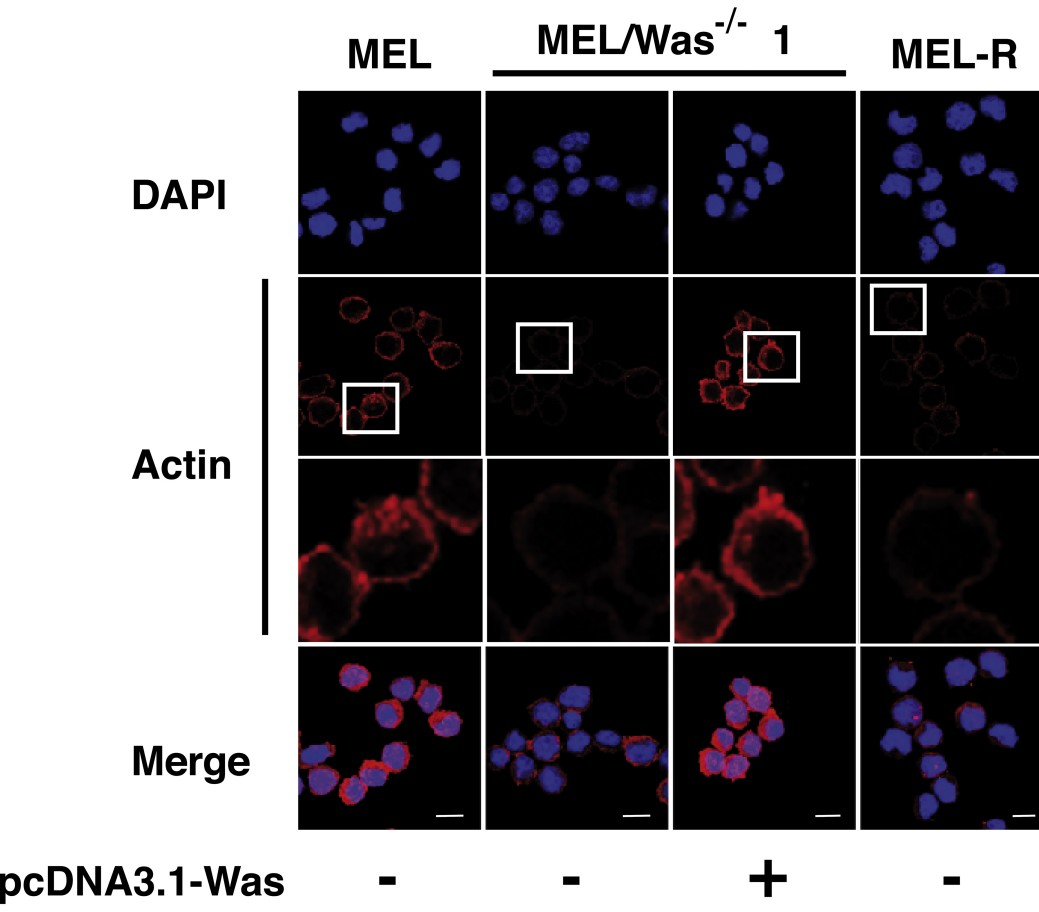

**Figure 5** **Deletion of *Was* provokes defects in the organization and polymerization of actin.** Confocal images showing actin stained with a mouse monoclonal anti-b-actin antibody (red). Nuclei were visualized with DAPI (blue). Forced expression of *Was* in MEL/*Was*$^{-/-}$ clone1 was performed by transient transfection with pcDNA3.1-*Was* (column 3). Magnified views indicated by white boxed areas are shown below second-row panels. Scale bar represents 10 mm.

activators, such as PU.1 (*Christie et al., 2015*; *Himmelmann et al., 1996*). We confirmed these results by confocal microscopy, observing a marked increase in the levels of Btk antibody staining in all three MEL-R clones overexpressing *Was*, and also Btk positivity in wild-type MEL and MEL/*Was*$^{-/-}$ cells (Fig. 8).

## DISCUSSION

Our previous work aimed to profile differentially-expressed genes between progenitor MEL cells and the derived cell line MEL-R, with induced resistance to differentiation (*Fernandez-Calleja et al., 2017*). We identified a group of genes comprising three main features: down-regulated in the resistant cell line, preferentially expressed in the hematopoietic lineage, and implicated in the actin cytoskeleton organization. Interestingly, several of these genes, for example, *Was*, *Btk* and *Rac2*, give rise to severe phenotypes when mutated in humans (*Ambruso et al., 2000*; *Bosticardo et al., 2009*; *Conley et al., 2009*). In the present

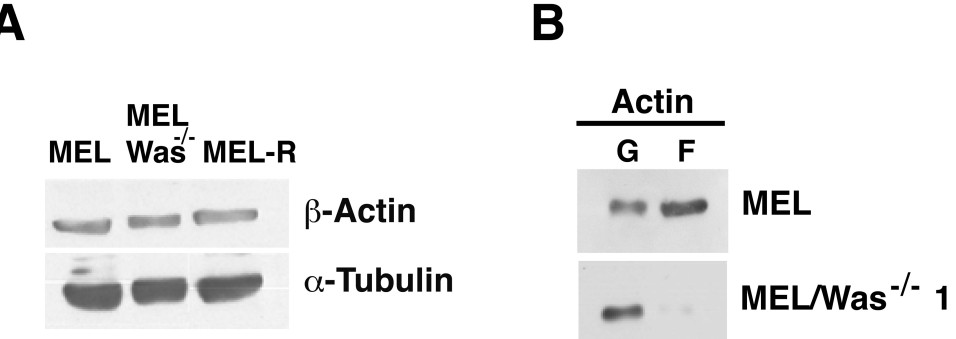

**Figure 6  G-/F-actin ratio is altered in MEL/$Was^{-/-}$ cells.** (A) Whole-cell lysates from MEL, MEL/Was$^{-/-}$ and MEL-R cells were analyzed by immunoblotting with an antibody against b-actin. α-tubulin was used as a loading control. (B) G-actin and F-actin from MEL and MEL-/$Was^{-/-}$, separated after ultracentrifugation, were immunoblotted and probed as in (A).

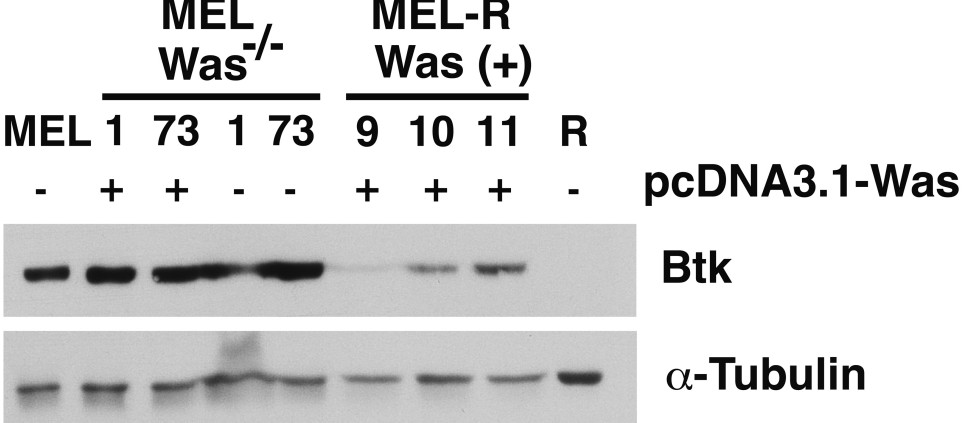

**Figure 7  Induction of *Was* expression in MEL-R cells stimulates Btk expression.** Immunoblotting of whole-cell lysates from MEL, MEL/Was$^{-/-}$ (clones 1 and 73), MEL-R/Was (+) and MEL-R cells with a mouse monoclonal anti-Btk-antibody. Ectopic expression of *Was* by transient transfection with the pcDNA3.1-Was vector marked (+) for clones MEL $Was^{-/-}$ 1 and 73 and MEL-R *Was* (+) 9, 10 and 11. α-tubulin was used as a protein loading control

study, we investigated the influence of *Was* expression on the organization of the actin cytoskeleton in murine erythroleukemia cells. A previous study by Symons and co-workers using transient transfection of WASp-expressing vectors demonstrated that WASp has a profound effect on actin polymerization in rat kidney epithelial, monkey COS7 kidney and Jurkat T cells (*Symons et al., 1996*). Furthermore, retrovirus-mediated expression of WASp was shown to reconstitute the actin cytoskeleton in human hematopoietic stem cells and myeloid derivative cells as well as in T and B cells and macrophages (*Dewey et al., 2006*) and references therein). Our results show that forced expression of WASp in the erythroid lineage MEL-R cell line helped to rebuild actin cytoskeleton organization. At the molecular level, we found that WASp overexpression induced the conversion of G-

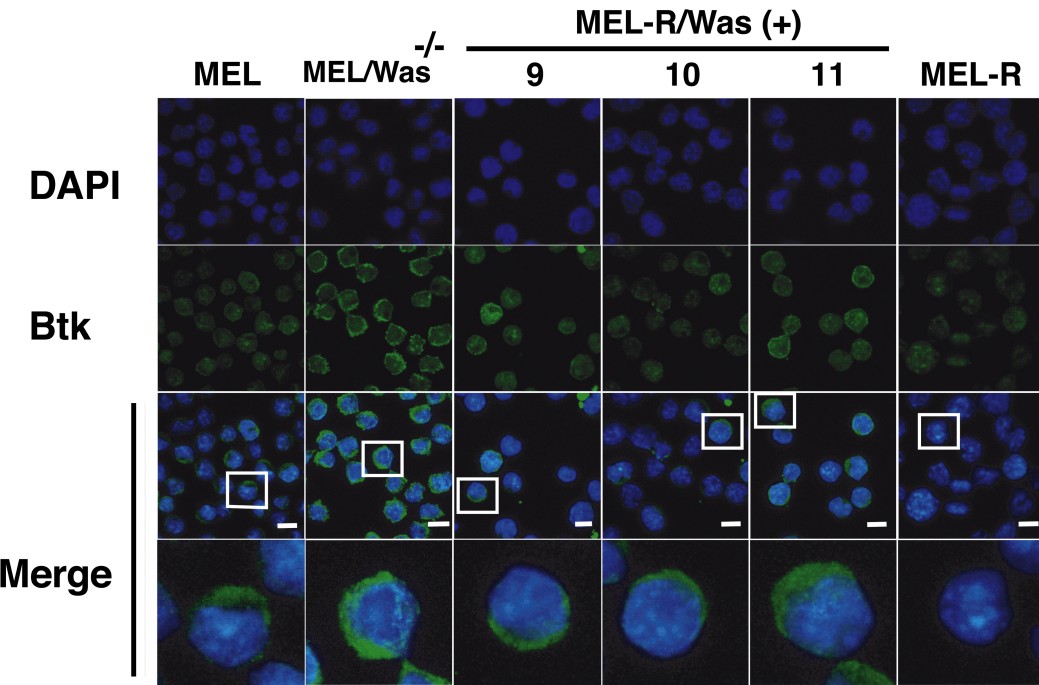

**Figure 8** **Induction of *Was* expression in MEL-R cells stimulates Btk expression.** Confocal immunofluorescence images of MEL, MEL/*Was*$^{-/-}$ (clone1), MEL-R/*Was* (+) (clones 9, 10 and 11) and MEL-R cells stained with an anti-Btk monoclonal antibody (green). Nuclear DNA was stained with DAPI (blue). Selected cells of each samples in white boxes areas are amplified below. Scale bar represents 10 mm.

to F-actin, which resulted in and increased concentration of F-actin. WASp expression is crucial for actin filament nucleation, a task that is carried out through the regulation of the Arp2/3 complex (*Machesky & Insall, 1998*). A classical view of actin homeostasis involves the ratio of G-/F-actin that is coordinated through signaling cascades (*Blanchoin et al., 2014*; *Burke et al., 2014*). The constant competition for a limited pool of actin monomers is tightly controlled by regulatory factors such as profilin-1, which in turn antagonizes WASp (*Rotty et al., 2015*; *Suarez et al., 2015*). By using CRISPR/Cas9 genome editing in MEL cells, we corroborated that the suppression of WASp has a deleterious effect on the actin cytoskeleton. Furthermore, the G-/F-actin ratio reflects a critical imbalance at the expense of the filamentous component. Taken together, our results suggest that WASp deficiency impedes correct cytoskeleton organization likely by blocking F-actin polymerization.

Phosphorylation of WASp is crucial for multiple cellular responses (*Blundell et al., 2009*). Btk is a non-receptor kinase that can phosphorylate WASp and relieve the autoinhibitory conformation that ultimately blocks actin polymerization. Btk specifically phosphorylates tyrosine 293 (Y293 in mice, Y291 in humans) within the GTPase binding domain, which triggers the destabilization of the autoinhibited conformation and facilitates binding of the Arp2/3 complex (*Blundell et al., 2009*). Sakuma and co-workers demonstrated that the interaction between the WASp N-terminal domain and the SH3 domain of Btk plays important roles in the lipopolysaccharide-TLR4 signaling cascade in macrophages (*Sakuma*

*et al., 2015*). Most of the studies concerning the interaction between WASp and Btk indicate that WASp is located downstream of Btk, as first suggested for B cell development (*Baba et al., 1999*). Our findings indicate that, directly or indirectly, WASp modulates the expression of Btk. Beyond WASp, other factors can act on Btk, as is the case of the transcription factor PU.1 (*Christie et al., 2015*; *Himmelmann et al., 1996*). PU.1 is expressed in MEL cells, whereas it is silenced in MEL-R cells. A possible reason why the expression of Btk was not altered in MEL/$Was^{-/-}$ cells might be that PU.1 has redundant activity. In MEL-R cells, however, the absence of both WASp and PU.1 would prevent the activation of Btk. Indeed, as we show here, restoration of WASp activates Btk. These seemingly contradictory findings suggest that the interactions governing WASp and Btk, and probably other members of the actin network, are complex.

Finally, what would be the importance of actin cytoskeleton-related proteins during erythropoiesis? MEL cells, and their MEL-R variants are derived from proerythroblasts infected with the Friend virus complex. When treated with inducers of differentiation, MEL cells can complete the differentiation program and reach the reticulocyte stage. Under specific conditions, when grown in the presence of a fibronectin matrix, a large proportion of MEL cells will complete enucleation (*Patel & Lodish, 1987*). MEL-R cells, however, are unable to grow on fibronectin-coated plates and cannot complete enucleation loss (*Fernandez-Nestosa, 2007*). Enucleation is a complex process that mirrors cytokinesis and, accordingly, the actin cytoskeleton and the multiple proteins that make it up have a fundamental role (*Konstantinidis et al., 2012*). The deregulation of these proteins could negatively influence and prevent enucleation.

## CONCLUSIONS

In the present work, we used CRISPR/Cas9 to delete the Was gene in erythroleukemia cells. We show that Was-deficient cells have a poor organization of the actin cytoskeleton that is accompanied by an imbalance in the ratio of monomeric G- and polymeric F-actin; a defect that is reversed by restoring Was expression. We also demonstrate that Was overexpression mediates the activation of another member of the actin cytoskeleton network, Bruton's tyrosine kinase. Overall, our results support the role of WASp as a key mediator of F-actin regulation and illustrate its importance in the erythroid lineage.

## ACKNOWLEDGEMENTS

We thank members of the ''Microtubule stabilizing agents'' group, J-Fernando Díaz and Daniel Lucena-Agell for their technical advice on ultracentrifugation and G-/F-actin-separation protocols. We acknowledge Mayte Dominguez and Gemma Rodriguez for their help with confocal images analysis and Alicia Bernabé for technical help.

### Funding

This work was supported by grants BFU2014 to Jorge B. Schvartzman, Pablo Hernández and Dora B. Krimer from the Ministerio de Economía y Competitividad of Spain, ICOOPB20224 from the National Spanish Research Council (CSIC) to Dora B Krimer and María-José Fernández-Nestosa, and PINV15-573 from CONACYT, Paraguay to María-José Fernández-Nestosa, and Jorge B Schvartzman. The funders had no role in study design, data collection and analysis, decision to publish, or preparation of the manuscript.

### Grant Disclosures

The following grant information was disclosed by the authors:
Ministerio de Economía y Competitividad of Spain: ICOOPB20224.
National Spanish Research Council (CSIC).
CONACYT.

### Competing Interests

Dora B. Krimer is an Academic Editor for PeerJ.

### Author Contributions

- Vanessa Fernández-Calleja conceived and designed the experiments, performed the experiments, analyzed the data, contributed reagents/materials/analysis tools, prepared figures and/or tables, authored or reviewed drafts of the paper, approved the final draft.
- María-José Fernández-Nestosa analyzed the data, contributed reagents/materials/analysis tools, authored or reviewed drafts of the paper, approved the final draft.
- Pablo Hernández and Jorge B. Schvartzman analyzed the data, authored or reviewed drafts of the paper, approved the final draft.
- Dora B. Krimer conceived and designed the experiments, analyzed the data, prepared figures and/or tables, authored or reviewed drafts of the paper, approved the final draft.

### Data Availability

Raw data is provided in the Supplemental Files.

### Supplemental Information

Supplemental information for this article can be found online at http://dx.doi.org/10.7717/peerj.6284#supplemental-information.

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
