# Peer review of "CRISPR/Cas9-mediated deletion of the Wiskott-Aldrich syndrome locus causes actin cytoskeleton disorganization in murine erythroleukemia cells"

_PeerJ, doi:10.7717/peerj.6284_

## Round 0.1 · original submission · Major Revisions

Dear Dr Krimer,

Your manuscript entitled "CRISPR/Cas9-mediated deletion of the Wiskott-Aldrich syndrome locus causes actin cytoskeleton disorganization in murine erythroleukemia cells" has now been seen by 2 referees. You will see from their comments below that while they find your work of interest, some important points are raised.

We are interested in the possibility of publishing your study in PeerJ, but would like to consider your response to these concerns (or explain satisfactorily why their comments are invalid) in the form of a revised manuscript before we make a final decision on publication. In particular when possible, please provide quantification and statistical analysis. However, I do believe that point 3 of reviewer 2 is beyond the scope of the paper.

We look forward to receiving your revised manuscript,

Yours sincerely,

Alexis Verger

·

Basic reporting

The manuscript is well organized and clearly written, including sufficient detail and references for the background material.

Experimental design

The manuscript describes a well defined set of experiments to study the effect of WAS gene deletion in a murine erythroleukemia cell line. The results are original and within the scope of PeerJ. Overall, the methods are described in adequate detail and the investigations are well performed. However, there is a lack of quantification and statistical evaluation of the results. The images and western blots should be quantified to enable statistical testing to rigorously support the conclusions of the study. If the experiments have been replicated enough times, this should not require additional replicates, simply quantitative analysis of the findings.

Validity of the findings

The results are convincing and the data are consistent with the conclusions, although as noted above there is a lack of quantification and statistical analysis that should be addressed. The discussion of the relevance of the observations are well stated and appropriate.

Additional comments

This manuscript is a welcome addition to the field. The one clarification that should be made is defining HMBA on line 185 of the Results section in addition to it's definition in the Methods section on line 106.

Reviewer 2 ·

Basic reporting

See below

Experimental design

See below

Validity of the findings

See below.

Additional comments

This manuscript describes a role for Wiskott-Aldrich Syndrome protein (WASp) in regulating actin cytoskeleton organization in mouse erythrolukemia (MEL) cells. Deletion of the Was gene in locus in MEL cells using CRISPR-Cas9 genome editing technology resulted in increased G actin/F actin ratio in biochemical assays. Consistently, weak phalloidin staining for F-actin was observed in Was knockout MEL cells. However, ectopic WASp expression in the Was knock out background successfully rescued the poorly organized actin phenotype. Furthermore, it was also shown that WASp overexpression results in Bruton’s Tyrosine kinase expression. Although the experiments are well designed with appropriate controls, the study is not very novel as the role of WASp in actin cytoskeleton regulation in a variety of cell types has been already well established. Further experiments demonstrating the role of WASp in various aspects of erythroblast differentiation are required. The authors need to address these concerns listed below.

Major points:

1. How does WASp deletion affect erythroblast differentiation? Does it impair enucleation or other organelle clearance from the cells?
2. It has been shown that WASp overexpression results in Bruton’s Tyrosine kinase expression. What are the downstream targets of this kinase in the erythroblasts and how does this upregulation of Bruton’s tyrosine kinase affect erythroblast differentiation?
3. WASp helps to regulate the assembly of Arp2/3 complex and thereby helps in dendritic branching of F-actin filaments. Are there any differences seen in the dendritic branching structure of the actin cytoskeleton in the Was knockout MEL cells? Are their differences in the rates of actin assembly and turnover? Live cell actin dynamics assay with fluorophore tagged-Lifeact could be carried out in the WT and knock out cell lines along with rescue experiments with WASp.

Additional point:

1. The authors should provide quantification and statistical analysis of the western blots and immunofluorescence images in the manuscript.

---

## Round 0.2 · Minor Revisions

Thank you for submitting a revised version of your manuscript and please accept my apologies for

the delay in coming back to you with a decision. It took unfortunately a bit longer than anticipated for one of the referees to return his/her report.
As you will see they both find that revisions significantly improved the manuscript. However, they both recommend that the quantification should be included as supplementary figures in the manuscript.

I would like to invite you to submit your revised manuscript
while addressing theses comments.

·

Basic reporting

No comment.

Experimental design

No comment.

Validity of the findings

No comment.

Additional comments

The authors are correct to acknowledge the inherent problems with quantifying western blots that use chemiluminescence and film for detection, the absolute values aren't reliable. However, quantification gives a better view of the reproducibility between experiments than simply telling us that the presented results are representative of independent replicates, even if the takeaway message is only about trends rather than absolute values. For this reason, I strongly recommend that the quantification of the western blots be included in the primary figures. Similarly, the quantification of the image analysis should also be included in the primary figures. There is little point to include these data in the rebuttal for reviewers only, it doesn't serve the scientific community at all. We all want to know how reproducible results are, not just to see examples.

Reviewer 2 ·

Basic reporting

Acceptable.

Experimental design

Acceptable.

Validity of the findings

I was pleased to see that the authors had quantified their western blot and immunofliuorescence data. However, the quantification of the western blot and immunofluorescence data has only been provided in two Supplemental figures that are labeled as for the Reviewers. In addition, the data in the bar graphs have not been analyzed for statistical significance, and no P values are determined. This data must be included in the main article and P values must be provided. This data MUST be presented in the manuscript proper to allow the readers to assess the validity of the authors conclusions.

The argument that western blots are non-linear and thus it is not useful to provide quantification is not valid. If the westerns performed by the authors are non-linear then none of that data is robust, and the authors would need to modify the protein loading of the samples and repeat the experiments under conditions where the blots are linear. However, I will assume that they are approximately in the linear range, and that the authors can perform the quantification, as in fact they have shown in the Supplemental files. The authors can of course discuss the issues with westerns, if they choose.

Additional comments

No further comments.

---

## Round 0.3 · accepted · Accept

Thank you for the revised version of the manuscript. The manuscript is now suitable for the publication in PeerJ.

#